# CoAst: Validation-Free Contribution Assessment for Federated Learning based on Cross-Round Valuation

## ABSTRACT

In the federated learning (FL) process, since the data held by each participant is different, it is necessary to figure out which participant has a higher contribution to the model performance. Effective contribution assessment can help motivate data owners to participate in the FL training. The research work in this field can be divided into two directions based on whether a validation dataset is required. Validation-based methods need to use representative validation data to measure the model accuracy, which is difficult to obtain in practical FL scenarios. Existing validation-free methods assess the contribution based on the parameters and gradients of local models and the global model in a single training round, which is easily compromised by the stochasticity of DL training. In this work, we propose CoAst, a practical method to assess the FL participants' contribution without access to any validation data. The core idea of CoAst involves two aspects: one is to only count the most important part of model parameters through a weights quantization, and the other is a cross-round valuation based on the similarity between the current local parameters and the global parameter updates in several subsequent communication rounds. Extensive experiments show that the assessment reliability of CoAst is comparable to existing validation-based methods and outperforms existing validation-free methods. We believe that CoAst will inspire the community to study a new FL paradigm with an inherent contribution assessment.

## 1 INTRODUCTION

With the development of deep learning (DL), the concept that "Data is the new oil" has gained more and more consensus among people [5]. The emerging remarkable capabilities demonstrated by large language models [18] further bring attention to the enormous value of collaborating on a large amount of data. To train DL models on data owned by different parties, collaborative learning techniques, represented by federated learning (FL) [7, 23], have been extensively studied. However, due to disparities in the data held by different participants, including variations in data quality and quantity, each participant's contribution to the performance of the FL model differs a lot. How to accurately evaluate the contribution of each participant is crucial for the fair distribution of rewards. This process benefits the promotion of data quality and creates incentives for data sharing [17].

There have been a line of research in the community on contribution assessment of participants in FL, which can be mainly divided into two categories, i.e., validation-based methods [2, 6, 19] and validation-free methods [10, 14, 21, 22]. The effectiveness of validation-based methods heavily relies on a representative validation dataset, which is used to evaluate the model performance. However, in real-world FL scenarios, obtaining the representative validation dataset that covers the distribution of all clients' data can be challenging. To overcome the limitations imposed by the validation dataset, validation-free methods are proposed to assess

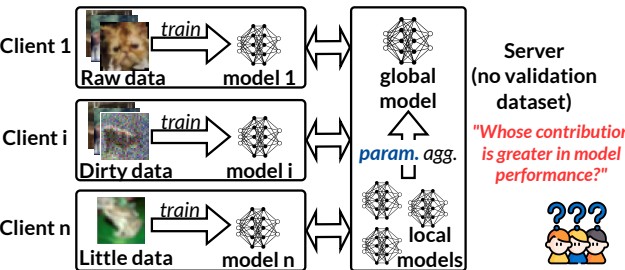

**Figure 1: Contribution assessment in FL scenarios.**

the contribution based on the statistical characteristics of model parameters. They estimate the parameters' correlation, information gain, and mutual information among local models (produced by clients) and the global model (aggregated by the server) to answer whose contribution is higher.

However, the existing validation-free efforts, detailed in the next section, only consider the models' parameters (or gradient) in a single training round. We refer to a training round as the process of the client updating and uploading the local model and getting the global model from the server after aggregation. Due to the difficulties, such as the stochastic nature of gradient descent and the uninterpretable nature of the DL model, the accuracy of these assessment methods can sometimes be seriously compromised. Performing validation-free assessment faces two challenges: *1)* the training process of DL models does not update parameters in a linear way, so only comparing the parameters of the local model in each round with the parameters of the final model will ignore the clients' contribution reflected in the iterative process of parameter updating. *2)* due to the parameter redundancy of DL models, not all parameters in the local model reflect the client's contribution to the performance improvement of the global model. Unimportant local parameters may comprise the assessment's effectiveness.

In this work, we propose CoAst, a validation-free FL contribution assessment method, performing client valuation in a *cross-round* way. The overview of CoAst is depicted in Figure 1. The core idea of the cross-round valuation is to evaluate a local model's contribution in a certain round (e.g., $t$) leveraging the parameter updates of global model in several subsequent rounds (e.g., rounds $\{t + 1, \ldots, t + k\}$). To cope with the interference of unimportant parameters, CoAst shares a similar idea with the ternary weights quantization [9, 20]. CoAst spotlights the most important parameters by neglecting the parameters with small updates and keeping only the sign of the parameters with a large update. It is worth noting that our assessment goal is to accurately and effectively measure each client's contribution to the FL model, rather than improve the FL model performace. As a result, CoAst neither affects the local training procedure nor changes the global models. Experimental results show that the assessment accuracy of CoAst is comparable to the validation-based methods and outperforms the existing validation-free methods. We

have submitted the source code in supplementary materials and will open-source the CoAst once the paper is accepted. We summarize the contribution as follows.

- We design a new validation-free contribution assessment method, CoAst, which can answer whose contribution is greater in a practical FL scenario without a validation dataset.
- We propose a cross-round assessment mechanism to consider the effect of the intermediate local models on the final trained model, and we utilize the ternary weight quantization to capture the parameters that contribute the most.
- Experimental results show that CoAst outperforms the SOTA validation-free methods in assessment effectiveness and achieves comparable -performance to validation-based methods.

The rest of the paper is organized as follows. In Section 2, we review the related works. In Section 3, we formulize the targeted scenario and problem. Section 4 presents the detailed design of our proposed CoAst. Then, we describe the experimental settings in Section 5 and provide the evaluation results in Section 6. The limitation of this work is discussed in Section 7. Finally, we conclude this paper in Section 8.

## 2 RELATED WORKS

**Validation-based methods.** The validation-based methods use a validation dataset to assess a client's contribution by evaluating its impact on the performance of the aggregated model. The leave-one-out is the most natural way to assess the value. It assesses the data value of one contributor by calculating the model performance change when the contributor is removed from the set of contributors. However, the leave-one-out is unfair to multiple similar and mutually substitutable contributors. Ruoxi *et al.* [6] use the Shapley value to assess data value in the FL scenario. They compute the marginal increase of the average accuracy of the model due to the addition of one data contributor. Guan *et al.* [15] extend the application scenarios of Shapley value-based solutions to FL scenarios. Zhenan *et al.* [2] apply the Shapley value-based solutions to vertical federated learning and improve the efficiency through approximation. Zhaoxuan *et al.* [19] allow for efficient data valuation without long-term model training. They assess the contribution through a domain-aware generalization bound, which is derived from the neural tangent kernel (NTK) theory. There are also research efforts to improve the system performance [11, 13, 16] and to analyze the fairness [25]. When the server uses this line of work to assess the clients' contribution, it requires a representative dataset which covers the distribution among all clients' data. However, in real-world FL scenarios, obtaining such a representative validation dataset is infeasible.

**Validation-free methods.** The validation-free methods use statistics of training data or the correlation among local and global parameters to value the clients. These works usually have some specific assumptions on the distribution of gradients, model parameters, or local training data. Therefore, they may face performance degradation in the real world when their assumptions are not satisfied. Xinyi *et al.* [22] propose a volume measure on the replication robustness, which assesses the contribution based on the diversity of training data. However, work [19] shows that this idea not only suffers from exploding volumes in high-dimensional inputs but also

entirely ignores the useful information in the validation dataset. Rachael *et al.* [14] measure the data value based on the information gains of the model parameters. They hold that the contributors with the highest value can reduce the uncertainty of the model parameters. However, when the distribution of data is complex, the accuracy of the information gains is biased. Xinyi *et al.* [21] use the gradient similarity to measure the data value of the contributors' combination by comparing the data of one combination of the contributors with the gradient similarity of the global FL model trained by all contributors. However, due to the randomness of the stochastic gradient descent and gradient pruning, the value assessed in some rounds may not accurately reflect the true value of the data, or even the value of high-value data is negative. Hongtao *et al.* [10] propose a test data-free data value evaluation based on the pairwise correlation among the models based on the statistical characteristics of the models. They assume that the parameters trained by different contributors share the same distribution, which may not be satisfied when the data is imbalanced and non-independent, and identically distributed.

## 3 PROBLEM OVERVIEW

### 3.1 Targeted Scenario

We assume all participants, including clients and the server, are honest and follow the agreed-on training protocol of FL. Due to differences in training data quality and quantity among clients, each client's contribution to overall model performance varies. The server can access the model parameters uploaded by each client, but it lacks a representative validation dataset. Each client delegates the server to evaluate the contribution of each client without offering validation data. Without loss of generality, we assume that each client is involved in all training rounds. The training data of each client is prepared before the FL training, and they will not add new data during the training process.

### 3.2 Problem Formalization

Consider an FL training procedure with one server and $N$ clients. The training procedure consists of $M$ training rounds. Recall that, in one training round, a client uploads its local parameters to the server once and receives the corresponding aggregated parameters (a.k.a. global parameters). After $M$ training rounds, the contribution of all clients is determined. We denote the ground-truth contribution of all clients as $P = \{p_1, p_2, \ldots, p_N\}$, where $p_i$ is the contribution of client $i$. The ranking of all clients' contribution is denoted as $R = \{r_1, r_2, \ldots, r_N\}$, and

$$r_i = |\{j | p_j \geq p_i\}|, \tag{1}$$

where $|\cdot|$ returns the number of elements in a set.

Our goal is to design a function $L$, which can measure how much each client improves the performance of the global model. That is,

$$\hat{P} = L\big(\Theta, \{\theta_i^t\}_{i \in [1,N], t \in [1,M]}\big), \tag{2}$$

where $\Theta$ denotes the parameters of the global model $\phi_\Theta$ in the last round, and $\theta_i^t$ denotes the parameters of the local model trained by client $i$ in round $t$. Then the predicted $\hat{R}$ can be calculated based on the $\hat{P}$ through Equation 1. The objective of function $L$ is to minimize

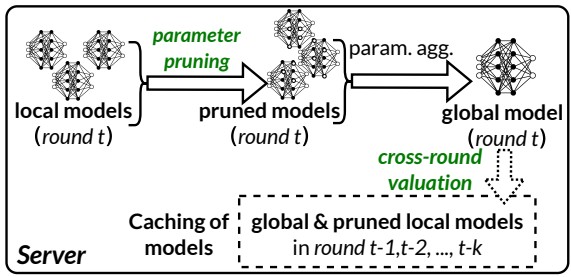

**Figure 2: The CoAst's workflow. Note that the local training procedure is unchanged.**

the distance of the predicted $\hat{R}$ and $R$, i.e.,

$$\min d(\hat{R}, R), \tag{3}$$

where $d(\cdot, \cdot)$ is the distance measurement function.

Due to the multi-round property of FL algorithm, the contribution score of each client in round $t$ (denoted as $\hat{P}^t$) can be naturally represented as:

$$\hat{P}_i^t = L(\Theta^t, \{\theta_i^t\}), \tag{4}$$

where $\Theta^t$ is the parameters of the global model aggregated in round $t$. So, we can reformulate the Equation 2 to

$$\hat{P} = \{\sum_{t=0}^{M} \hat{P}_i^t | i \in [1, N]\}. \tag{5}$$

## 4 COAST DESIGN

### 4.1 Design Overview

We propose two key techniques to address the challenges introduced in the introduction section. *First*, we design a cross-round valuation mechanism. Due to stochastic gradient descent, gradient pruning, and parameter regularization, the parameter updates in a certain round (e.g., round $t$) may not accurately reflect the true value of the client, and even high-value clients may be assigned negative contributions. Fortunately, the FL training process minimizes the optimization objective and improves the model's accuracy, which means that the global parameter updates have a positive contribution in most training rounds. It allows us to value the client $i$ in round $t$ with global parameters of subsequent several rounds.

*Second*, we borrow ideas from efforts on model compression and quantization to filter out those unimportant parameters. Binary weight quantization [4] is proposed for the efficiency of computation and storage and demonstrates that retaining only the sign of parameters during model updates can still reach acceptable accuracy. Then, the ternary weight quantization [9] sets unimportant parameter updates to zero on top of binary weight quantization and reaches comparable accuracy to full precision training. It shows that by setting a threshold, parameter updates that contribute minimally to the model performance can be effectively filtered out. Thus, we apply the idea of ternary weight quantization to the global model aggregation and then value the client's contribution with the remaining important parameters.

We demonstrate the workflow of CoAst in Figure 2. The contribution assessment is transparent to clients, and there is nothing to

---

**Algorithm 1:** The parameter pruning algorithm of CoAst.

**Data:** All clients' parameters at the $t$ epoch: $\theta_1^t, \theta_2^t, \ldots, \theta_N^t$;
Pruning Ratio: $r \in (0, 100]$;
Normalization hyperparameter: $\alpha$;
Aggregated Parameter at the $(t-1)$-th round: $\Theta^{t-1}$.
**Result:** Pruned Local Parameters : $\tilde{\theta}_1^t, \tilde{\theta}_2^t, \ldots, \tilde{\theta}_N^t$.

1 **begin**
2    $\tilde{\theta}_1^t, \tilde{\theta}_2^t, \ldots, \tilde{\theta}_N^t$ = zero_init$(\theta_1^t, \theta_2^t, \ldots, \theta_N^t)$
     // The zero_init function returns multiple zero tensors with the same shapes as inputs
3    $\Delta_1, \Delta_2, \ldots, \Delta_N = \Theta^{t-1} - \theta_1^t, \Theta^{t-1} - \theta_2^t, \ldots, \Theta^{t-1} - \theta_N^t$
4    **for** $i$ *in* $\{1, ..., N\}$ **do**
5      **for** $j$ *in* $\{1, ..., l\}$ **do**
6        $I = \text{argsort}(\Delta_i[j])$
         // $\Delta_i[j]$ denotes $j$-th layer's parameter updates
         // argsort$(\cdot)$ returns indices by the value in descending order
7        $I_\Delta = [I_1, I_2, ..., I_{\lfloor |I| \cdot r\% \rfloor}]$
         // $I_m$ denotes the $m$-th element of the variable $I$
8        **for** $h$ *in* $I_\Delta$ **do**
9          $\tilde{\theta}_i^t[j][h] = \Theta^{t-1}[j][h] + \alpha \cdot \text{sgn}(\Delta_i[j][h])$
           // The sgn$(\cdot)$ is the sign function
           // $\theta_i^t[j][h]$ denotes the $h$-th element at $j$-th layer of the parameter update $\Delta_i$
10    **return** $\tilde{\theta}_1^t, \tilde{\theta}_2^t, \ldots, \tilde{\theta}_N^t$

---

change during the local training procedure. On the server, when receiving the local parameters trained in round $t$, CoAst first prunes the parameters by their importance through the ternary weight quantization, then performs the aggregation on the pruned models. After aggregation, we use cross-round valuation to measure the contributions of local models. In the next part, we will detail these two key designs.

### 4.2 Parameter Pruning

Without loss of generality, we use the training process of round $t$ as an example to detail the algorithm design. $N$ clients first locally train the local models of round $t$ based on the global parameters of round $t-1$, denoted as $\Theta^{t-1}$. Then, all clients upload local parameters of round $t$ to the server, which are denoted as $\{\theta_i^t\}_{i \in \{1, ..., N\}}$. In the following procedure, clients do nothing but wait for the aggregated global parameters of round $t$, denoted as $\Theta^t$, to continue their local training in the next round.

Once all local parameters $\{\theta_i^t\}_{i \in \{1, ..., N\}}$ are received, CoAst calculates the local updates of round $t$, denoted as $\{\Delta_i^t\}_{i \in \{1, ..., N\}}$, where

$$\Delta_i^t = \theta_i^t - \Theta^{t-1}. \tag{6}$$

Then CoAst performs parameter pruning according to their importance, similar to the idea of model quantization such as binarization and ternarization. Considering the structural and functional differences between layers of DL models, CoAst quantifies the parameter importance in a layer-wise manner. Here we denote the parameter updates in each layer as $\Delta_i := [\delta_1, \delta_2, \ldots, \delta_l]$ where $l$ refers to the number of layers. Take the $j$-th layer as an example. CoAst first

calculates the $r$-th percentile of $|\delta_j|$, denoted as $\delta_j^r$, where $r$ is a hyperparameter controlling the pruning rate. Then, CoAst prunes the parameters by clipping the parameter updates. For each element of $\delta_j$, denoted as $u$, it is clipped as:

$$\tilde{u} = \begin{cases} 1 & \text{if } u > \delta_j^r \\ -1 & \text{if } u < -\delta_j^r \\ 0 & \text{otherwise} \end{cases}. \tag{7}$$

It means we regard the elements of $\delta_j$ whose absolute values are greater than $\delta_j^r$ as important and only keep their signs, while the remaining elements are pruned to 0. Given the clipped parameter updates $\tilde{\Delta}_i^t$, the pruned local parameters $\tilde{\theta}_i^t$ can be calculated as

$$\tilde{\theta}_i^t = \Theta^{t-1} + \alpha \cdot \tilde{\Delta}_i^t, \tag{8}$$

where $\alpha$ is a hyperparameter for normalization. We report the whole parameter pruning procedure in Algorithm 1.

## 4.3 Cross-round Valuation

In each round, the contribution of the model is valued by the global model aggregated of the next $k$ rounds, and it also values the local models of the last $k$ rounds. Recall that the parameters of the global model are the average of the pruned local parameters (Equation 8), which are calculated by adding the sign of the parameter update. Therefore, the global parameters $\Theta^t$ is calculated by

$$\Theta^t = \frac{1}{N} \cdot \sum_{i=1}^{N} \tilde{\theta}_i^t = \Theta^{t-1} + \frac{\alpha}{N} \cdot \sum_{i=1}^{N} \tilde{\Delta}_i^t. \tag{9}$$

Therefore, in round $t$, the parameter update of the global model denoted as $U^t := \Theta^t - \Theta^{t-1}$, is proportional to the sum of the pruned local updates, i.e.,

$$U^t \propto \sum_{i=1}^{N} \tilde{\Delta}_i^t. \tag{10}$$

Recall that the value of element $\tilde{u} \in \Delta_i'$ belongs to $\{-1, 0, 1\}$. That is, the $U^t$ (Equation 10) is the normalized voting result, which indicates, for each element $b \in \Theta^{t-1}$, how many clients believe that its value should be increased by $\frac{\alpha}{N}$, and how many clients believe that it should be decreased by $\frac{\alpha}{N}$.

After $k$ rounds following the $t$-th round, we obtain the parameters $\Theta^{t+k}$, which can be calculated by

$$\Theta^{t+k} = \Theta^t + \sum_{e=t+1}^{t+k} U^e. \tag{11}$$

In the following part, we denote $\sum_{e=t+1}^{t+k} U^e$ as $U^{(t,k)}$ for simplification. Since the global parameters are obtained by averaging all local parameters, we can regard the parameters of any client as a correction to the sum of parameters of the other $N-1$ clients. Without loss of generality, for any client $i$, we reformulate Equation 11 to

$$\Theta^{t+k} = \frac{1}{N} \cdot \sum_{j \in \{1,\ldots,N\} \setminus \{i\}} \theta_j^t + \frac{1}{N} \cdot \theta_i^t + U^{(t,k)}. \tag{12}$$

We assume that the model's performance is improved after the $k$ rounds. That is, among all clients, if a client's local model of round $t$, i.e., $\theta_i^t$, during the parameter aggregation process is more similar to the subsequent global parameter updates, i.e., $U^{(t,k)}$, then

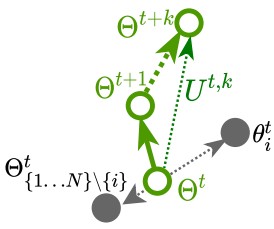

**Figure 3: Core idea of the cross-round valuation. The $\theta_i^t$, which is consistent with the improvement direction of the model in several rounds, i.e., $U^{(t,k)}$, has a larger contribution.**

the contribution of this client in this training round is greater. We demonstrate this idea in Figure 3. Clients will recalculate the local models based on the last global model, so the choice of $k$ should not be too large. Otherwise, the update direction represented by $U^{(t,k)}$ may not capture the details of stochastic gradient updates and thus cannot indicate the contribution in one training round.

Therefore, the contribution of client $i$ can be measured by the similarity between $\theta_i^t$ and $U^{(t,k)}$. Here, we use Signed Cosine similarity to measure the similarity because the $\theta_i^t$ and $U^{(t,k)}$ are local parameters and global updates, respectively. Note that Signed Cosine similarity is sensitive to the sign information of vectors and can better reflect the directional relationship between vectors. Due to the parameter pruning, the model updates can be considered as the sign of each parameter's update (Equation 10). That is, in our design, the sign of these updates is more important than their magnitude.

Note that $\theta$ and $U^{(t,k)}$ share the same shape, and we assume that they can be indexed through $h$. CoAst calculates the client's contribution in round $t$ by

$$\hat{p}_i^t = \sum_{h=1}^{|\Theta|} \text{sgn}(\theta_i^t[h]) \cdot \text{sgn}(U^{(t,k)}[h]), \tag{13}$$

where $\text{sgn}(\cdot)$ is the function to indicate the sign of the value.

# 5 IMPLEMENTATION

## 5.1 Dataset Settings

We evaluate the CoAst's performance on three datasets, i.e., CIFAR-10 [8], CIFAR-100 [8], and STL-10 [1]. We randomly partition the training dataset among each participant. We assume that by randomly and evenly partitioning these three datasets according to the number of clients, several datasets with the same data valuation can be obtained. Therefore, if a group of clients uses these partitioned datasets for training, the contribution of these clients is the same. We have set up four scenarios to mimic the contribution differences caused by data quality and quantity differences. $N$ in the following settings denotes the number of clients.

*5.1.1 Setting 1: Different quantity.* Assuming that randomly partitioned datasets share a similar distribution, the more samples in the training dataset, the higher the contribution to the model accuracy. In this scenario, we prepare datasets with different contributions by randomly assigning different numbers of samples to each client. Let the number of clients be $N$ and the size of the

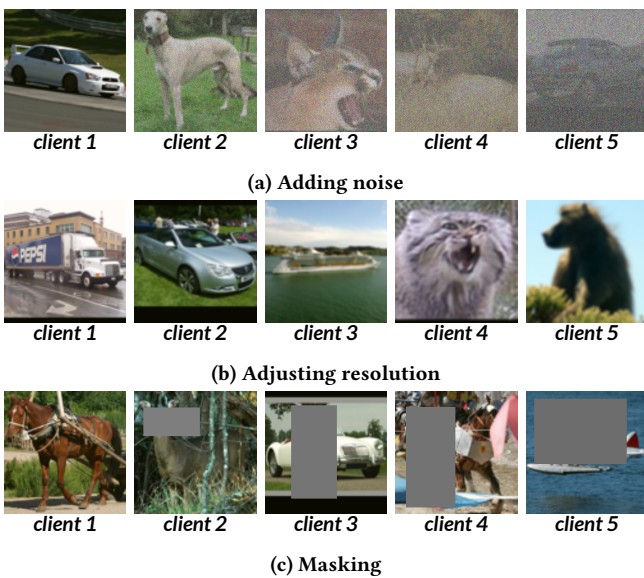

**(a) Adding noise**

**(b) Adjusting resolution**

**(c) Masking**

**Figure 4: Training samples to demonstrate our dataset settings. Samples are randomly selected from STL-10.**

training dataset be $|X_{\text{train}}|$. Then, the size of the dataset for the $i$-th client is $D_i = 1 - 0.5 \cdot \frac{i}{N}|X_{\text{train}}|$.

*5.1.2 Setting 2: Adding noise.* In this scenario, we prepare the training datasets of different qualities for clients by adding Gaussian noise of different intensities. We first randomly and evenly partition the dataset according to the number of clients. Then we perform the Gaussian noise to the dataset of client $i$ with a mean of $\mu_i$ and standard deviation of $\sigma_i$, where $i \in \{1, \ldots, N\}$. We set the mean and variance of Gaussian noise decrease linearly, i.e., $\mu_i = 0.01 \cdot i, \sigma_i = 0.625 \cdot \frac{i}{N}$. We report some data samples in Figure 4a.

*5.1.3 Setting 3: Adjusting resolution.* In this scenario, we mimic the data of different quality by adjusting the resolution of the training data through Gaussian blur. Different degrees of Gaussian blur can be achieved by setting kernels of different sizes and variances. We first randomly and evenly partition the dataset according to the number of clients. Then, we use different degrees of Gaussian blur to preprocess the training data. Let the sequence of kernel sizes and standard deviation be $s_i$ and $\sigma_i$, where $i \in \{1, \ldots, N\}$. We select a linearly decreasing sequence of kernel sizes and standard deviation, i.e., $s_i = 2 \cdot i + 1, \sigma_i = 0.4 \cdot i + 1$. We report some data samples in Figure 4b.

*5.1.4 Setting 4: Masking.* In this scenario, we prepare the dataset with different quality by adding a mask on training data. The content covered by the mask is set to 0. We first randomly and evenly partition the dataset according to the number of clients. Then, for each client, we randomly mask a part of the image. The area of the mask covers $r_i\%$ of the image for client $i$, and its position is randomly generated. The $r_i$ is a random number between $l_i$ and $u_i$, where $l_i = 0.5 \cdot \frac{i}{N}, u_i = 0.75 \cdot \frac{i}{N}$. We report some data samples in Figure 4c.

## 5.2 Implementation Details

We perform experiments with Pytorch on a server with two A100 (80G) GPU cards. We use three model architectures, i.e., TinyResNet, ConvNet [24], and ResNet-4. The TinyResNet consists of a convolution layer, whose weight shape is $3 \times 7 \times 7 \times 64$, and a ResBlock [3], whose kernel size is $3 \times 3$ and output channel is 64. The ResNet-4 consists of 4 ResBlocks, whose kernel size is $3 \times 3$ and output channel is 64, 128, 128, and 128. In our setting, we use 1 central server and 5 clients, (i.e., $N = 5$). We initialize the learning rate to 0.01 and gradually decrease it as the training progresses. We use the FedAvg method to aggregate the local models.

## 5.3 Baseline Methods

We use three baseline methods. The validation-based method [15], denoted as *baseline 1*, measures the accuracy of the aggregated models w/ or w/o one local model with the validation dataset to calculate the contribution. Although baseline 1 is computationally time-consuming, it has the highest accuracy among Shapley value-based solutions by exhaustively considering all possible cases. We implement two SOTA validation-free methods, i.e., Fed-PCA [10] and CGSV [21], as the *baseline 2* and *baseline 3*.

## 5.4 Metrics

To measure the accuracy of the contribution assessment, we use the Spearman correlation coefficient [12] as the distance measurement function $d$ in Equation 3. The Spearman correlation coefficient is good at measuring the degree to which the ranks of the two variables are associated with each other. And it also is used in related works [17, 21]. Formally, $R$ and $\hat{R}$ are two sequences of length $n$, which denote the ground-truth order and predicted order of the clients' contribution, i.e., $R = [o_1, o_2, \ldots, o_n]$, $\hat{R} = [\hat{o}_1, \hat{o}_2, \ldots, \hat{o}_n]$. We calculate the Spearman correlation coefficient ($\rho$) through the:

$$\rho = 1 - \frac{6}{n(n^2 - 1)} \sum_{i=1}^{n} \|o_i - \hat{o}_i\|_2. \tag{14}$$

The $\rho$ ranges from -1 to 1, where -1 indicates a perfectly negative correlation, i.e., sequences $R$ and $\hat{R}$ are in reverse order, 0 indicates no correlation, i.e., random guessing, and 1 indicates a perfectly positive correlation.

## 6 EVALUATION

We measure the performance of our CoAst in four configurations with three datasets and three model architectures. **Configuration 1** is CIFAR-10 and TinyResNet; **Configuration 2** is CIFAR-100 and TinyResNet; **Configuration 3** is STL-10 and ResNet-4; **Configuration 4** is CIFAR-100 and ConvNet.

## 6.1 Overall Performance

In the CoAst's experiments, for hyperparameters in Algorithm 1, we set $r$ to 10, $\alpha$ to 0.02, and $N$ to 5. We set the $k$ in Equation 13 to 2. We report the overall performance in Table 1. SV, PCA, and CGSV represent baseline 1, baseline 2, and baseline 3, respectively.

The average accuracy of baseline 1 and CoAst are 0.855 and 0.9, which means that our CoAst is comparable to that of baseline 1,

| C. | Setting 1: Quantity | | | | Setting 2: Noise | | | | Setting 3: Resolution | | | | Setting 4: Mask | | | |
|---|---|---|---|---|---|---|---|---|---|---|---|---|---|---|---|---|
| | SV | PCA | CGSV | **Ours** | SV | PCA | CGSV | **Ours** | SV | PCA | CGSV | **Ours** | SV | FP | CGSV | **Ours** |
| 1 | 0.60 | -1.00 | **1.00** | **1.00** | 1.00 | 0.30 | 0.30 | **1.00** | 1.00 | 0.60 | 0.30 | **1.00** | 1.00 | 0.00 | -0.70 | **1.00** |
| 2 | 0.90 | -0.10 | **1.00** | **1.00** | 1.00 | 0.10 | 0.60 | **1.00** | 1.00 | 0.10 | -0.30 | **0.90** | 1.00 | 0.10 | -1.00 | **1.00** |
| 3 | 0.70 | 0.00 | **1.00** | **1.00** | 0.90 | 0.30 | 0.30 | **1.00** | 1.00 | **0.70** | 0.40 | 0.20 | 0.10 | -0.30 | 0.40 | **0.60** |
| 4 | 0.90 | -0.10 | **1.00** | **1.00** | 1.00 | -1.00 | -0.10 | **1.00** | 1.00 | -0.40 | **0.70** | **0.70** | 1.00 | 0.10 | -0.30 | **1.00** |
| **A.** | 0.78 | -0.30 | **1.00** | **1.00** | 0.98 | -0.08 | 0.28 | **1.00** | 1.00 | 0.25 | 0.28 | **0.70** | 0.78 | -0.03 | -0.40 | **0.90** |

**Table 1: Overall performance of CoAst. SV [15] represents baseline 1, a validation-based method. Note that some efforts, in these years, optimize the system performance by estimating the Shapley value rather than improving the assessment accuracy (Section 2). Therefore, we use the classical SV [15] to compare accuracy. PCA [10] and CGSV [21] represent baseline 2 and baseline 3. Our CoAst (denoted as O). (C. is short for configuration, and A. is short for Average.)**

which is a validation-based method. CoAst even outperforms baseline 1 by 0.22, 0.02, and 0.12 in setting 1, setting 2, and setting 4. Our CoAst's performance outperforms the SOTA validation-free methods, i.e., Fed-PCA (baseline 2) and CGSV (baseline 3), in almost all cases. On average, our CoAst outperforms Fed-PCA by 0.94 and CGSV by 0.52 in all cases. The poor performance of Fed-PCA is because the model architecture used in our experiment is too complex to perform precise probability analysis. Experimental results demonstrate the CoAst's effectiveness and robustness in different cases.

## 6.2 Ablation Study

*6.2.1 $k$ in cross-round valuation.* We explore how $k$ affects the performance of CoAst. We perform the experiment with different $k$ values, which means that different numbers of global updates are used to assess the local models' contribution. We report the results in Table 2. In different experimental settings, CoAst with $k = 5$ reaches the best performance, and the performance of CoAst with $k = 2$ is comparable to that of CoAst with $k = 5$. The small and large $k$ values, i.e., $k = 1$ and $k = 10$, have relatively poor performance. This is because a small $k$ value may still let the contribution assessment be affected by stochasticity, while a large value of $k$ may make CoAst lack attention to the stochastic gradient descent process. We recommend setting $k$ to 2 or 5 when using our CoAst in practical use.

*6.2.2 Parameter pruning.* Recall that the parameter pruning consists of a parameter update clipping procedure (Line 9 in Algorithm 1) and a top r% update selection procedure (Line 7 in Algorithm 1). To measure the effect of parameter pruning, we perform the following contrast experiments with different settings, which are as follows.

(1) Ours. $r = 10$, w/ update clipping.
(2) Exp1. $r = 20$, w/ update clipping.
(3) Exp2. $r = 10$, w/o update clipping.
(4) Exp3. $r = 100$, w/o update clipping. (No parameter pruning.)

We report the results of these contrast experiments in Table 3. By comparing the results of ours and Exp1, we can conclude that increasing the proportion of parameter selection leads to a slight decrease in performance, which is likely due to the noise introduced by the selected parameters. Our CoAst's average performance is

0.40625 and 0.25625 higher than Exp2 and Exp3, respectively, indicating that the design of parameter pruning effectively ensures the accuracy of contribution assessment.

*6.2.3 Update clipping strategy.* In the CoAst, we use a hyperparameter $\alpha$ (Equation 8) to normalize the local parameter update in each round. Here we explore how the hyperparameter $\alpha$ value affects the contribution assessment effectiveness. We perform three experiments with CIFAR-10 and TinyResNet. We set $M$ to 100, $N$ to 5, and $r$ to 10. In the first two experiments, we set the value of $\alpha$ to 0.01 and 0.02, respectively. In the third experiment, we use an adaptive clipping strategy, where we set $\alpha$ as the average value of the selected r% parameters (i.e., 10) of each layer. We report the results of these three experiments in Table 4. By comparing the experimental results of the first two experiments, it can be seen that the choice of hyperparameters has little impact on the assessment performance. However, in the third experiment, the assessment performance decreases. This is because clipping the parameter update to different values interferes with the assessment process. In our design, we aim to quantize all parameter updates and assess the contribution based on the direction of the local parameters and global parameter updates.

*6.2.4 Client number.* Here we explore how the number of clients, i.e., $N$, affects the stability of CoAst. We experiment with CIFAR-100 and TinyResNet and set $k$ to 10 and $r$ to 10. We report the experimental results in Table 5. When the number of clients increased from 5 to 10, the performance change of Fed-PCA averaged 0.4775, the performance change of CGSV averaged 0.38, and the accuracy change of our method averaged 0.07. When the number of clients is 5, our CoAst outperforms Fed-PCA and CGSV by an average of 1.025 and 0.5225, respectively. When the number of clients is 10, our CoAst outperforms Fed-PCA and CGSV by an average of 0.775 and 0.795, respectively. The experimental results fully demonstrate the stability of our method with respect to changes in the number of clients.

## 7 DISCUSSION AND LIMITATION

Recall that in the procedure of parameter aggregation, CoAst prunes local parameters according to their importance. Although weight quantization methods are often used in practical FL scenarios to reduce network bandwidth, they tend to incurring performance degradation of the model as well as slower convergence speed. Thus the convergence time of our CoAst is slightly longer than that of

| C. | Setting 1: Quantity | | | | Setting 2: Noise | | | | Setting 3: Resolution | | | | Setting 4: Mask | | | |
|---|---|---|---|---|---|---|---|---|---|---|---|---|---|---|---|---|
| | k=1 | k=2 | k=5 | k=10 | k=1 | k=2 | k=5 | k=10 | k=1 | k=2 | k=5 | k=10 | k=1 | k=2 | k=5 | k=10 |
| 1 | 1.00 | 1.00 | 1.00 | 1.00 | 1.00 | 1.00 | 1.00 | 1.00 | 1.00 | 1.00 | 1.00 | 1.00 | 0.90 | 1.00 | 1.00 | 1.00 |
| 2 | 1.00 | 1.00 | 1.00 | 1.00 | 1.00 | 1.00 | 1.00 | 1.00 | 1.00 | 0.90 | 0.90 | 0.90 | 1.00 | 1.00 | 1.00 | 1.00 |
| 3 | 0.80 | 1.00 | 1.00 | 0.90 | 1.00 | 1.00 | 1.00 | 1.00 | 0.10 | 0.20 | 0.60 | 0.50 | 0.60 | 0.60 | 0.60 | -0.60 |
| 4 | 1.00 | 1.00 | 1.00 | 1.00 | 1.00 | 1.00 | 1.00 | 1.00 | 0.70 | 0.70 | 0.70 | 0.70 | 0.90 | 1.00 | 1.00 | 1.00 |
| A. | 0.95 | **1.00** | **1.00** | 0.98 | **1.00** | **1.00** | **1.00** | **1.00** | 0.70 | 0.70 | **0.80** | 0.78 | 0.85 | **0.90** | **0.90** | 0.60 |

**Table 2: The effect of the number of subsequent global updates, i.e., $k$, on the contribution assessment's performance. (C. is short for configuration, and A. is short for Average.)**

| C. | Setting 1: Quantity | | | | Setting 2: Noise | | | | Setting 3: Resolution | | | | Setting 4: Mask | | | |
|---|---|---|---|---|---|---|---|---|---|---|---|---|---|---|---|---|
| | O | Exp1 | Exp2 | Exp3 | O | Exp1 | Exp2 | Exp3 | O | Exp1 | Exp2 | Exp3 | O | Exp1 | Exp2 | Exp3 |
| 1 | 1.00 | 1.00 | 1.00 | 1.00 | 1.00 | 1.00 | 1.00 | 1.00 | 1.00 | 0.10 | -0.70 | 0.30 | 1.00 | 1.00 | 1.00 | 1.00 |
| 2 | 1.00 | 1.00 | 1.00 | 1.00 | 1.00 | 1.00 | 1.00 | 1.00 | 0.90 | -0.10 | -0.10 | 0.60 | 1.00 | 1.00 | 1.00 | 1.00 |
| 3 | 1.00 | 1.00 | -0.30 | 0.40 | 1.00 | 0.70 | 0.40 | 0.40 | 0.20 | -0.40 | -0.40 | -0.40 | 0.60 | 0.70 | -0.70 | -0.70 |
| 4 | 1.00 | 1.00 | 1.00 | 1.00 | 1.00 | 1.00 | 1.00 | 1.00 | 0.70 | 0.40 | 0.70 | 0.70 | 1.00 | 1.00 | 1.00 | 1.00 |
| A. | **1.00** | **1.00** | 0.68 | 0.85 | **1.00** | 0.93 | 0.85 | 0.85 | **0.70** | 0.00 | -0.13 | 0.30 | 0.90 | **0.93** | 0.58 | 0.58 |

**Table 3: Experimental results of exploring how $r$ and update clipping affects the assessment performance. (C. is short for configuration, and A. is short for Average.)**

| $\alpha$ | Quantity | Noise | Resolution | Mask |
|---|---|---|---|---|
| 0.01 | 1.00 | 1.00 | 0.90 | 1.00 |
| 0.02 | 1.00 | 1.00 | 1.00 | 1.00 |
| avg | 1.00 | 1.00 | 0.40 | 1.00 |

**Table 4: The effect of update clipping strategy on contribution assessment. We perform the experiment with CIFAR-10 and TinyResNet. We set $N$ to 5, and $M$ to 100.**

| C. | Typical | Ours | | |
|---|---|---|---|---|
| | M=100 | M=100 | M=150 | M=200 |
| 1 | **0.7963** | 0.7732 | 0.7816 | 0.7880 |
| 2 | **0.4931** | 0.4456 | 0.4555 | 0.4555 |
| 3 | 0.5776 | 0.5186 | 0.5590 | **0.5876** |
| 4 | 0.5767 | **0.5883** | **0.5906** | **0.5906** |

**Table 6: The performance of the FL model trained in a typical way and our methods. M represents the number of training rounds. (C. is short for configuration.)**

| #Client | Quantity | | | Noise | | |
|---|---|---|---|---|---|---|
| | PCA | CGSV | **Ours** | PCA | CGSV | **Ours** |
| 5 | -1.00 | **1.00** | **1.00** | 0.30 | 0.30 | **1.00** |
| 10 | 0.30 | **0.99** | **0.99** | 0.86 | 0.88 | **0.99** |

| #Client | Resolution | | | Mask | | |
|---|---|---|---|---|---|---|
| | PCA | CGSV | **Ours** | FP | CGSV | **Ours** |
| 5 | 0.60 | 0.30 | **1.00** | 0.00 | -0.70 | **1.00** |
| 10 | 0.61 | -0.37 | **0.82** | -0.14 | -0.96 | **0.92** |

**Table 5: Experimental results of how the number of clients affects the performance of contribution assessment.**

the typical method, i.e., without applying any quantization. We report the model performance trained through the typical method and ours in Table 6. The model trained through the typical method converges after 100 rounds (i.e., $M = 100$), while the model trained in our method requires 200 rounds (i.e., $M = 200$) to converge. As can be seen, the performance of the model trained through CoAst outperforms that of the model trained through the typical method at the 100th round only in one case. When the model converges, the performance of the model trained through CoAst exceeds the model trained in the typical scenario in two cases. However, experiments in Section 6.2.2 show that weight quantization is important to

the accuracy of the assessment since quantization mitigates the negative impact of redundant model parameters on the assessment. As future work, we will draw on research ideas in the direction of model quantization to improve the model performance in accuracy and convergence.

## 8  CONCLUSION

In this work, we propose a validation-free contribution assessment, CoAst, for the FL scenario. Compared with existing efforts, it greatly improves the contribution assessment performance under different dataset settings by introducing two key designs: parameter pruning and cross-round valuation. Comprehensive evaluations showed that our CoAst outperforms existing methods on different dataset settings and different models. We believe that CoAst will inspire the data valuation design in other scenarios in the future.

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
