# OpenReview forum: "CoAst: Validation-Free Contribution Assessment for Federated Learning based on Cross-Round Valuation"
_acmmm.org/ACMMM/2024/Conference — MM2024 Poster_

### Official Review · Reviewer_bsFC · 2024-05-25

**Rating:** 4
**Confidence:** 3

**Summary:**

This paper proposes a method to evaluate the user contribution in federated learning. The method requires no validation data. It measures the contribution by the similarity of model updates. Experiment results demonstrate the performance of the proposed method in 4 scenarios (data quantity, noise, resolution, masking), using the Shapley ratio as the ground truth.

**Strengths:**

1.	The paper is well-written and easy to understand.
2.	The proposed method is reasonable, as the similarity with the global model update certainly reflects the contribution of the single client’s update.
3.	The claim is supported by the experiment results.

**Limitations:**

1.	The proposed method is somewhat trivial, since it is only a similarity computation along with sparsification.
2.	The ‘low-quality’ data used in the experiments are man-made according to some simple rules. It is obvious that those ‘tampered’ data will make the model update deviate the average. So I think it would be better if the authors could conduct some experiments on natural data, e.g., partitioning the dataset into high-quality data and low-quality data.
3.	It is also good to test the method on noisy-labeled data, i.e., some of the clients get a portion of mislabeled samples.
4.	It seems currently, the proposed method is just to compute the contribution scores, however, the model performance is not affected. It would be great to have a discussion about how to use those contribution scores.

**Suitability:**

2

---

### Official Review · Reviewer_mCwS · 2024-05-25

**Rating:** 4
**Confidence:** 1

**Summary:**

This paper studies the problem of validation-free contribution evaluation in federated learning. To achieve this, the authors utilize weight quantization and cross-round parameter comparison. Experiments are conducted on real-world datasets.

**Strengths:**

1. The target problem is interesting, and has a significant impact on the field of federated learning.
2. The paper is well-organized and easy to follow.

**Limitations:**

1. How to determine k in practical application?
2. How to eliminate the differences between the pruned model and the original model during evaluation?

**Suitability:**

2

---

### Official Review · Reviewer_V6YK · 2024-05-27

**Rating:** 3
**Confidence:** 2

**Summary:**

This paper propose a novel contribution assess method in FL without the need of validation dataset.

**Strengths:**

1. The pruning parameter methods are intuitive.

**Limitations:**

1. The result tables are a little bit confusing. Why the metrics are close to 1? Does this metric make sense on the assessment of contribution?
2. Can the idea works on the Non-IID scenarios?
3. The overall assessment algorithm framework is encouraged to be provided.

**Suitability:**

1

---

### Official Review · Reviewer_bAz6 · 2024-05-31

**Rating:** 4
**Confidence:** 3

**Summary:**

This paper explores contribution assessment problem in federated setting. The authors proposed a new validation-free contribution assessment that can measure contributions of each client without a validation dataset. Experiments show that the proposed method is effective compared to existing works.

**Strengths:**

1. The paper studies the validation-free contribution assessment problem, which is an interesting issue in federated learning.

2. The paper is well organized.  The implementation details and experiments are clear and easy to read.

**Limitations:**

1. The experiment was implemented with only five clients and the same level of heterogeneity. How does the model perform in real-world situations that are often encountered in federated learning, such as heterogeneity (through Dirichlet distribution [1]) and more clients?

2. It seems to me that the baseline SV outperforms the proposed method in experiment setting 3. It's better for authors to explain this result.

3. The number of clients in the experiment was increased to 10. It's very common for contemporary federated learning to have hundreds of clients or more. Can the proposed algorithm effectively scale up to handle such a scenario?

4. Why is baseline SV not compared in ablation experiments about client numbers?

[1] Ganghua Wang, Ali Payani, Myungjin Lee, and Ramana Kompella. Mitigating group bias in federated learning: Beyond local fairness. arXiv preprint arXiv:2305.09931, 2023.

**Suitability:**

2

---

### Meta-Review · Area_Chair_FDL7 · 2024-06-26

**Recommendation:** Accept (Poster)
**Confidence:** 4

**Metareview:**

The reviewers think this paper interesting, intuitive, well organised, but still have some limitations. In the first round, this paper got 3BA anf 1BR. After the rebuttal, the BR is changed into BA, so this paper got 4BA. This score is OK for a poster.